# Relationship between Prosocial Behaviours and Addiction Problems: A Systematic Review

**DOI:** 10.3390/healthcare10010074

**Published:** 2021-12-31

**Authors:** Javier Esparza-Reig, Manuel Martí-Vilar, César Merino-Soto, Alfredo García-Casique

**Affiliations:** 1Department of Basic Psychology, Faculty of Psychology and Speech Therapy, University of Valencia, 46800 Valencia, Spain; javieresparzareig@gmail.com; 2Research Institute of Psychology, University of San Martín de Porres, Lima 15102, Peru; 3Department of Psychology, Faculty of Psychology, Federico Villarreal National University, Lima 15088, Peru; agarcia@unfv.edu.pe

**Keywords:** prosocial behaviour, addiction, gambling, substance abuse, positive psychology

## Abstract

The relationship of addiction problems with other pathologies or with different problematic factors has often been studied by psychology. Positive psychology is also currently approaching to these problems and their relationship with positive factors, such as prosocial behaviours. The purpose of this research is to carry out a systematic review of the scientific literature that has studied this relationship from 1900 to 2020. After the screening process with the inclusion and exclusion criteria, a total of 15 articles were selected. The main characteristics found in this relationship and the problems or limitations of investigations that have found relationships other than the mainstream, which show a negative relationship between prosocial behaviours and addiction problems, are discussed.

## 1. Introduction

Research on addiction problems has often been focused on their relationship with other psychological disorders by the field of psychology. Depression, one of the psychological disorders affecting more people today, has been shown to be related to both substance addiction [1] and addictive behaviours, such as social media addiction [2]. In the case of anxiety, another widespread mental disorder, the same relationship as in the case of depression has been found with both addictive behaviours [3] and substance addiction [1].

Another relationship that has been widely studied is the one that addiction problems keep with what is considered negative personality traits. Research on the relationship between addictions and antisocial behaviours has found that problematic alcohol consumption in adolescence is associated with antisocial behaviours in adulthood [4]; the use of other addictive substances, such as cannabis and tobacco, also appears to be directly related to these behaviours [5]. In the case of addictive behaviours, research suggests a similar relationship. For example, Pivetta, Harkin, Billieux, Kanjo, and Kuss [6] reported that smartphone addiction was directly related to antisocial behaviours. Other traits, such as low self-esteem [7] or lack of empathy [8,9], are directly related to addiction problems, too.

The concept of positive psychology emerged in the late 1990s [10] to name a stream of psychology that sought to find, develop, and evaluate interventions focused on enhan-cing positive aspects and human well-being rather than focusing on pathologies, negative aspects, or problems [11]. This stream arises from the basis that psychopathologies and well-being are independent constructs but closely related [12]. Positive psychology aims to enhance aspects such as optimism, personal growth, forgiveness, enjoying good things, or being grateful for positive experiences, among others [13].

Prosocial behaviour is closely linked to positive psychology [14] and is one of the aspects to be enhanced to improve people’s well-being [15]. These behaviours aim to help others and are beneficial to them, helping to avoid antisocial behaviours or aggressive attitudes [16,17]. As a consequence of this characterization of prosocial behaviour, it is apparently incompatible that prosocial behaviours coexist with behaviours that decrease the expression of prosociality. This is the case with addictive behaviours. Addictive behaviour becomes the main focus of a person’s life, excluding other activities, or begins to harm the individual and others physically, mentally, or socially.

The study of addictions from the positive psychology point of view, specifically the relationship of addictions with prosocial behaviours, lacks extensive research, especially when compared with the previously indicated research of pathologies, negative personality traits, or interpersonal relationships. The absence of extensive research is not clear, but it is possibly driven by intuitive reasoning that both must covariate negatively due to the apparent incompatibility between them. However, this gap is an opportunity to show patterns of scientific findings that may have theoretical and applied value.

Existing research finds a two-way relationship in which prosocial behaviour acts as a protective factor against the development of addiction problems [18,19]. On the other hand, people with addiction problems tend to carry out fewer prosocial behaviours than those who do not have these problems [20,21]. It is important to note that many of these investigations focus on adolescents and not adults.

The aim of this research is to carry out a synthesis of the existing information in the existing literature on the relationship between prosocial behaviours addiction problems in adults, whether they are addictions to substances or behaviours, to know the current state of the research about this theme.

## 2. Methods

A systematic review (SR) was carried out after the recommendations of the PRISMA guideline [22] for this type of research. Appendix A includes a checklist following the PRISMA guide. In addition, a bibliometric analysis of the documents selected after the screening process was carried out. 

First, a search in the Tripdatabase and Cochrane databases was performed to verify that there were no previous SR or meta-analysis that had already investigated this relationship. This was followed by a search in the Web of Science (WoS; main collection), Scopus, PsycInfo, and Dialnet databases.

The purpose was to search for “prosocial behaviour”. However, this term appears in the scientific literature with many synonyms, such as “prosociality”. The word prosocial* was included in the search equation since it is the root of the word prosocial. In screening, eligibility, and inclusion, all those articles that evaluated prosocial behaviour were eliminated.

The search equation used in WoS was Prosocial* AND (Gambl* OR Addict* OR “Substance Abuse” OR Drug*), and it was therefore adapted for the rest of the databases. In the case of Dialnet, the search equation was written in Spanish and was the following: Prosocial* AND (Ludopati* OR Adic* OR Abuso OR droga*). Finally, a complementary search was done in Google Scholar to include grey literature and avoid the possible publication bias.

The search was made in titles, abstracts, and keywords, and no restriction was included for years. Thus, it covers from 1900 to July 2021 (however, no documents prior to 1991 were found). Only empirical research articles were selected. After removing duplicate records, the total number was 815.

These 815 selected articles were analysed in two steps. In the first one, only the title and abstract of the documents were read. The complete document was read in the second one. The screening process was carried out on the basis of the following criteria:(a)Documents should be written in English or Spanish, excluding any other language.(b)Documents should be empirical research articles with quantitative method. Books, doctoral theses, editorials, lectures, clinical case studies, literature reviews, and other research with non quantitative method were discarded.(c)Investigations should be conducted on humans; animal investigations were excluded.(d)The addiction problems analysed should be present in adults. Articles that studied the effect of prosocial behaviours in adolescence or childhood on addiction problems in adulthood were also selected. Investigations that focused on the problems of addiction in minors were discarded.(e)Documents that analysed the relationship between prosocial behaviours and addictions were selected. Those that did not analyse this relationship were discarded. We also ruled out research that analysed prosocial jobs, prosocial attachments, or prosocial environments instead of prosocial behaviours.(f)All documents focusing on the neurobiological basis of addictions and the effect of drugs or treatments without focusing on the relationship between prosocial behaviours and addiction problems were discarded.

The selection and screening process of documents is detailed in Figure 1.

## 3. Results

A total of 15 documents that met the criteria were found to be part of the review. Table 1 shows the main characteristics of the 15 documents sorted alphabetically by authorship.

First, all the articles analysed were published in the last 20 years: 80% in the last 10 years and 53.3% of them originating within the last five years (taking into account that at the time of this review, the year 2021 has not yet been completed, and the number of articles published during that year may increase). Moreover, 40% (*n* = 6) of the investigations were carried out in the USA; 40% were in Spain, China, and Switzerland (*n* = 2 for each country); and the remaining 20% were done in England, the Netherlands, and Canada (*n* = 1 for each country).

With regard to the characteristics of the studies, 86.6% (*n* = 13) of them applied validated instruments to measure the participants’ prosocial behaviours, while 60% (*n* = 9) applied validated instruments or performed biological tests to measure substance use or addiction problems. Only two studies [23,24] included groups of participants with diagnosed addiction problems, representing 13.3% of the total articles analysed in the review.

**Table 1 healthcare-10-00074-t001:** Information of the articles selected for the review.

Authors (Year) and Country	Studied Addiction and Aims	Sample	Design	Method	Results
Carlo, Crockett, Wilkinson, and Beal (2011); USA [25]	Tobacco, cannabis and alcohol; To investigate the potential longitudinal effect of prosocial behaviours during adolescence on substance use in young adults	*n* = 357 (final sample) (200 women); 23.11 years (0.94); general population	Longitudinal	5-year longitudinal study; to measure substance use, a 6-point Likert item was applied to collect the consumption of tobacco, alcohol, and marijuana; to measure prosocial behaviours, the Primary Prevention Awareness, Attitudes, and Usage Scale (PPAAUS; Swisher et al., 1984 [26]) was applied	Prosocial behaviour in adolescence is negatively correlated with the use of tobacco, cannabis, and alcohol
Collins and Freeman (2013); England [27]	Video game; To investigate how pathological video-gaming may be related to social personality traits, social capital, and prosocial behaviours	*n* = 416 (156 women); 27.23 years (8.48)	Cross-sectional	To measure the frequency of video-gaming use, the hours per week were asked; to measure pathological video-gaming use, the short version of the Game Addiction Scale (GA; Lemmens, Valkenburg, and Peter, 2009 [28]) was used; to measure prosocial behaviours, the Prosocial Tendencies Measure (PTM; Carlo and Randall, 2002 [29]) was applied	Prosocial behaviour is not related to video-gaming frequency or pathological video-gaming
Davis et al. (2016); USA [30]	Alcohol and cannabis; To explore the longitudinal bi-directional relationship between prosocial behaviours and substance use in a sample of college athletes	*n* = 187; they were all women; 19.87 years (1.25) (at the beginning of the study) at the last data collection *n* = 70; university students	Cross-sectional	To measure prosocial behaviours, the PTM (Carlo and Randall, 2002) was applied; for dangerous alcohol consumption, the Alcohol Use Disorders Identification Test (AUDIT; Saunders et al., 1993 [31]) was applied; for cannabis use, a 7-point Likert item was applied to collect its consumption during the last 30 days	Public prosocial behaviour is a positive predictor of alcohol consumption (self-oriented for the approval of others, not altruistic); cannabis use is a negative predictor of anonymous prosocial behaviour (altruistic, other-oriented)
Esparza-Reig (2020); Spain [32]	Gambling addiction; To deepen the study of prosocial behaviour in young university students to better understand its functioning and its effects on the development of gambling addiction problems	*n* = 258 (153 women); 20.95 years (3.19); university students	Cross-sectional	To measure gambling addiction problems, the South Oaks Gambling Screen (SOGS; Lesieur and Blume, 1987 [33]) was applied; to measure prosocial behaviour, the Prosociality Scale (Caprara et al., 2005 [34]) was applied	Prosocial behaviour is a protective factor against gambling addiction problems
Esparza-Reig, Martí-Vilar, and González-Sala (2021); Spain [35]	Gambling and alcohol addiction; To check whether, among other factors, prosocial behaviour acts as a protective factor against gambling and alcohol addiction problems	*n* = 258 (153 women); 20.95 years (3.19); university students	Cross-sectional	To measure gambling addiction problems, the South Oaks Gambling Screen (SOGS; Lesieur and Blume, 1987 [33]) was applied; for dangerous alcohol consumption, the Alcohol Use Disorders Identification Test (AUDIT; Saunders et al., 1993 [31]) was applied; to measure prosocial behaviour, the Prosociality Scale (Caprara et al., 2005 [34]) was applied	Prosocial behaviour is a protective factor against both gambling addiction problems and problematic alcohol use
Fenzel (2005); USA [36]	Alcohol; To examine a set of personality, environment, and risk and protective factors that are expected to be related to the frequency of binge drinking and alcohol-related problems in university students	*n* = 686 (472 women); 19.6 years (1.2); university students (general population)	Cross-sectional	To measure the frequency of binge drinking in the last two weeks, an item was applied, and a series of items of 3 alternatives that measured whether alcohol consumption had caused them different problems; to measure prosocial behaviours, the number of behaviours of this type carried out by the participants was consulted	Prosocial behaviour is a protective factor against the frequency of binge drinking and the problems derived from this behaviour
Groves, Gentile, Tapscott, and Lynch (2015); USA [37]	Video game; To test empirically the correlates and predictive validity of pathological video-gaming based on DSM-style criteria for pathological gambling	(Only research study 2) *n* = 504 (197 women); it does not indicate age, but they were undergraduate university adults; general population	Cross-sectional	For pathological gambling, 9 dichotomous items from the General Media Habits Questionnaire (GMHQ; Fisher, 1994 [38]) were used, which are adaptations of the DSM-IV criteria for pathological gambling; to measure prosocial behaviours, an item from the Social Interaction Survey (Linder, Crick, and Collins, 2002 [39]) was used	There are no differences (they are not correlated) between pathological and non pathological players in prosocial behaviour
Guo et al. (2021); China [40]	Internet; To analyse the relationship between different types of emotional and behavioural problems and Internet addiction problems	*n* = 30,581 young adults with an average age of 19.9 years (1.6)	Cross-sectional	To assess prosocial behaviour, the Strengths and Difficulties Questionnaire (Goodman, 1997 [41]) was used; to assess problematic Internet use, the Young’s Internet Addiction Test (IAT; Young, 1998 [42]) was used	Prosocial behaviour reduces the risk of problematic Internet use
Lemmens, Valkenburg, and Gentile (2015); Netherlands [43]	Video games; To analyse the reliability and validity of 4 instruments (short and long version/polytomous and dichotomous version) for measuring Internet Gaming Disorder, based on the 9 criteria of the DSM-5	*n* = 1251 (630 women); 24.8 years (8.1); general population	Cross-sectional	To measure Internet Gambling Disorder, a dichotomous scale was used (in two versions, one with 27 items and another one with 9 items); 5 prosocial items of the Strengths and Difficulties Questionnaire (Goodman, 1997 [41]) were used	Video-gaming addiction negatively correlates with prosocial behaviour in both samples, and with the different versions of the Internet Gaming Disorder assessment instrument
		*n* = 1193 (615 women); 24.4 years (7.6); general population	Cross-sectional	To measure Internet Gambling Disorder, a 6-point Likert-type scale was used (in two versions, one with 27 items and another one with 9 items); 5 prosocial items of the Strengths and Difficulties Questionnaire (Goodman, 1997 [41]) were used	
Moilanen and Lynn (2019); USA [44]	Substances of abuse; To analyse if “helicopter parenting” in emerging adulthood is related to adjustment outcomes (social competence, prosocial behaviour, depression, substance use, and lifetime criminality), and whether any associations are mediated by personal mastery and/or self-regulation	*n* = 302 (196 women); 21.57 years (1.9); general population	Cross-sectional	To assess substance use, participants were asked when they last used different substances of abuse; to assess prosocial behaviour, the PPAAUS prosocial behaviour subscale (Swisher et al., 1984 [26]) was applied	Prosocial behaviour do not correlate with substance abuse
Quigley and Maggi (2014); Canada [45]	Tobacco, alcohol, and cannabis; To compare the associations found between aggressive behaviours and substance use, with those found between prosocial behaviours and substance use	*n* = 1305 tobacco consumers. No descriptive data are provided for the sample in relation to sex and age	Longitudinal	Longitudinal study. Both substance use and prosocial behaviours are measured trough self-reports	Tobacco and alcohol are not related to prosocial behaviour in childhood; Prosocial behaviour during childhood is a positive predictor of cannabis use
		*n* = 1218 alcohol consumers. No descriptive data are provided for the sample in relation to sex and age *n* = 1292 cannabis consumers. No descriptive data are provided for the sample in relation to sex and age	Cross-sectional		
Schnakenberg and Lysaker (2019); USA [23]	Cannabis; To characterize the influence of lifetime cannabis use on emotional experience in prolonged psychosis and the influence of the interaction of cannabis use and emotional expression on social function	Group with addiction problems: *n* = 36 (3 women) participants with regular cannabis use in the last 3 or more years; 49.97 years (6.7); participants with schizophrenia Control group: *n* = 35 (9 women) participants without regular cannabis use in the last 3 or more years; 49.26 years (10.06); participants with schizophrenia	Cross-sectional	To measure cannabis use, the ASI (McLellan et al., 1980 [46]) was used; to measure prosocial behaviours, the Social Functioning Scale (SFS; Birchwood et al., 1990 [47]) was applied	Participants who regularly smoke cannabis show less prosocial behaviours than non-smokers
Sun et al. (2021); China [48]	Smartphone addiction; To investigate psychiatric symptoms during the COVID-19 quarantine among university students in China	*n* = 1912 university student; 20.28 years of average age (2.1)	Cross-sectional	To measure prosocial behaviours, items adapted from the Empathic Responding to SARS Scale (Lee-Baggley et al., 2004 [49]) and the Prosociality Scale (Caprara et al., 2005 [34]) were used; to measure smartphone addiction, an item that measured the time of use was applied	They found a negative correlation between prosocial behaviour and screen time
Tomei, Studer, and Gmel (2021); Switzerland [50]	Addiction to gambling, video-gaming, alcohol, cannabis, and nicotine; To examine the relationship between prosociality and different types of addictions in a sample of young adults	*n* = 5654 young adult men with an average age of 21.34 years (1.27)	Cross-sectional	To assess prosocial behaviours, the Prosociality Scale (Caprara et al., 2005 [34]) was applied; to measure addiction to gambling and alcohol, the DSM-5 diagnostic criteria were applied; to measure video-gaming addiction, the abbreviated version of the GA (Lemmens et al., 2009 [28]) was applied; to measure cannabis addiction, the Cannabis Use Disorder Identification Test (Adamson and Sellman, 2003 [51]) was applied	Participants with problems of addiction to gambling, video-gaming, and alcohol presented less prosocial behaviour than participants without addiction problems. In the case of tobacco and cannabis, no differences were found. This relationship is especially significant in the case of addictive behaviours
Vonmoos et al. (2019); Switzerland [24]	Cocaine; To examine the relationship between the change in cocaine use and the development of sociocognitive functioning and symptoms of type B personality disorders over the course of a year	*n* = 48 healthy controls (16 women); 30.3 years (8.9) *n* = 19 chronic cocaine users who increased use for 1 year (3 women); 31.5 years (9.4) (at the time of starting the study) *n* = 19 chronic cocaine users who decreased use for 1 year (5 women); 31.4 years (8.3) (at the time of starting the study)	Longitudinal	1-year longitudinal study; cocaine use was measured at first and after one year by blood concentration (19 participants increased use and 19 participants decreased it); prosocial behaviour was measured with the Distribution/Dictator Game (Charness and Rabin, 2002 [52]) at first and after one year	The group that decreased consumption and control was closer in their scores in prosocial behaviour after one year; those that increased the use of cocaine maintained the distance in the levels of prosocial behaviour with the control group

With regard to the addictions studied, the 15 articles analysed 29 relationships between a certain addiction and prosocial behaviour because some articles investigate the relationship with several addictions or the same relationship in different samples. Of these 29, 58.62% (*n* = 17) focused on the relationship between prosocial behaviours and substance addiction problems and 41.37% (*n* = 12) on the relationship between prosocial behaviour and addictive behaviours. In the case of addictive behaviours, the most studied (*n* = 7) was the addiction to video games, representing 24.13% of the relationships studied, while 13.79% (*n* = 4) analysed the relationship between prosocial behaviour and gambling. On the other hand, accordingly to the substance of abuse, the most studied were the addiction to alcohol (*n* = 7) and to cannabis (*n* = 5), with 24.13% and 17.24%, respectively. The next most studied addictive substance was tobacco (*n* = 3), and finally, 3.44% analysed the relationship with cocaine (*n* = 1). In the case of Moilanen and Lynn [44], the relationship between prosocial behaviour and substance abuse was studied, but no specific substance was specified.

Regarding the results of the relationships studied in isolation for each substance or addictive behaviour with prosocial behaviour, it was found that, in the case of video-gaming addiction, 6.89% of the relationships investigated (*n* = 2) found that this addictive behaviour correlated negatively with prosocial behaviour [43]. Another 10.34% did not find any relationship between problematic video-gaming and prosocial behaviour [27,37]. The relationship between prosocial behaviour and gambling was only analysed once [32], finding that both correlated negatively, with prosocial behaviour being a protective factor against gambling problems.

In the case of the relationship between cannabis use and prosocial behaviours, 10.34% (*n* = 3) of the results show that both variables are negatively related, while 3.44% (*n* = 1) reflected a positive relationship, and another 3.44% (*n* = 1) found no significant differences [50]. In particular, the results showed that prosocial behaviour during adolescence acts as a protective factor against problematic cannabis consumption in adulthood [25] and that cannabis users showed less prosocial behaviour in adulthood [23,30]. On the other hand, prosocial behaviour in childhood would lead to increased cannabis use in adulthood [45].

Overall, 6.89% (*n* = 2) of the research results show that prosocial behaviour during adolescence [25] and during adulthood [36] functions as a protective factor against problematic alcohol consumption in adulthood. Of the remaining results, 13.79% (*n* = 4) found no relationship between prosocial behaviour in childhood [45] and anonymous prosocial behaviour, e.g., altruistic and focused on others without seeking a reward [36], and problem alcohol use in adults. Davis et al. [30] found a positive relationship between alcohol consumption and prosocial behaviour, but in this case, it was a type of public behaviour focused on what others would think, which would not fit into the altruistic approach of prosocial behaviour.

Regarding research on the relationship of prosocial behaviour and tobacco use, 3.44% (*n* = 1) found that prosocial behaviours during adolescence acted as a protective factor against tobacco consumption in adulthood [25]. Of the remaining articles, 6.89% (*n* = 2) found no relationship between tobacco use and prosocial behaviour [45].

In the case of cocaine, only one study analysed the link between cocaine use and prosocial behaviour [24] and found that both correlated negatively so that the people who consumed the most had greater deficits in their prosocial behaviour. Finally, Moilanen and Lynn [44] found no relationship between prosocial behaviour and substance abuse, without specifying which substances were involved.

When making no distinctions between the specific addiction studied, 20.68% (*n* = 6) of the results in the investigation between addictive behaviours and prosocial behaviour found a negative relationship. Of this proportion, 17.24% (*n* = 5) found that both variables correlate negatively, and the remaining 13.79% (*n* = 4) found that prosocial behaviours were a protective factor against addiction problems. In the case of substance addictions, 31.03% (*n* = 9) found a negative relationship with prosocial behaviours, 6.89% (*n* = 2) found a positive relationship, and the remaining 20.68% (*n* = 6) did not find a relationship. Of the 31.03% that found a negative relationship, 6.89% (*n* = 2) identified prosocial behaviours as a protective factor against substance addiction problems, while the remaining 3.44% (*n* = 1) identified substance addiction problems as a predictor of lower levels of prosocial behaviour.

When a distinction between addictive behaviours and substance addiction was not made, the 51.72% of the results (*n* = 15) found a negative relationship between addiction problems and prosocial behaviours, 6.89% (*n* = 2) a positive relationship, and 34.48% (*n* = 10) found no relationship. Of the 51.72% of results with a negative relationship, 6.89% (*n* = 2) did not analyse if one of the two exerted as predictor of the other, 20.68% (*n* = 6) placed addiction problems as predictors of fewer prosocial behaviours, and 24.13% placed prosocial behaviours as a protective factor against addiction problems.

## 4. Discussion

The aim of this research was to carry out a systematic review of the literature that has studied the relationship between prosocial behaviours and problems of addiction in adults either to substance addiction or addictive behaviour. The last aim was to know the current status of the investigation with regard to this relationship.

First, the results show that this relationship has not been very analysed over the years, but it has begun to gain relevance in recent years (especially in the last decade). The majority of the research is concentrated in USA, but it has also been studied in more countries.

On the other hand, despite the fact that most of the research used validated tests or medical analysis to measure the levels of prosocial behaviour or addiction problems (5.12% and 33.33% of the research, respectively), there is still a significant percentage of studies that does not use them, which is something to take into account for the results’ interpretation. In addition, it is noteworthy that only 20% of researches include participants with diagnosed addiction problems, while most of them are focused on a population without specific problems, which is also a limit of the results obtained.

Regarding the type of addiction, 55.17% studied substance addictions, and 44.82% focused on addictive behaviours, which can be considered quite balanced. Among addictive substances, studies are centred mainly on alcohol and cannabis, accounting for 37.93% of research, while in addictive behaviours, the most studied is the video-gaming addiction, with 24.13% of the total.

Focusing on the relationship between prosocial and addictive behaviour, the results mainly give back a negative relationship between both variables. However, 13.79% (*n* = 4) of the research does not find any relationship. It should be noted that one of the two investigations that found this lack of relationship [37] did not apply any instrument to assess the prosocial behaviour, and it was limited to one item. The other study that found no relationship [27] points out in the limitations that they did not analyse if video game players did it in isolation or with friends, so they recommend taking this aspect into account when interpreting the results of their research.

On the other hand, in the case of the analysis of the relationship between prosocial behaviours and substance addiction, it is also reflected that there is a negative relationship between the two variables regardless of the substance of abuse. However, there are studies that have found that this relationship is positive (6.89% of the total) or have found that there is no such relationship (20.68%). When analysing the characteristics of these articles, it should be noted that Quigley and Maggi [45] found a positive relationship, but they studied the consequences of prosocial behaviours during childhood on the problems of cannabis use in adulthood, and they did not apply any instrument or validated test to measure the variables (all data are self-reported). In addition, these same authors found that prosocial behaviour in childhood was not related to alcohol and tobacco consumption in adulthood, and therefore, the same limitations already set out on this investigation should be taken into account when interpreting these results.

With regard to the direction of the relationship between prosocial behaviours and addiction problems, 13.79% of the research argues that prosocial behaviours are a protective factor against problems of addiction to gambling, while 20.68% argues that addiction problems are what cause deficiencies in prosocial behaviours. These results are not contradictory since it may be the case that prosocial behaviours truly are a protective factor but, in turn, develop addiction problems that generate negative effects on behaviours towards others.

### 4.1. Limitations

We must highlight some limitations. First, and which is possibly the main limitation of this systematic review, is the number of investigations analysed, which decreases its capacity for generalization. Second, most of the research was done in the USA, which skews the results toward English-speaking populations. Third, the use of valid measuring instruments in only 38.46% of the relationships weakens the results of studies where the validity of the measurement is induced or unknown. Fourth, correlational studies allow us to recognise relationships, associations, or simultaneous presence of variables, but they do not allow us to point out causes and effects between addictions and prosocial behaviour. Fifth, the size of the selected studies was generally small, which does introduce significant variations due to the size of the sampling error. Finally, it is considered important to point out that it was not possible to assess the quality of the studies that were included in the review due to the fact that there were only a limited number.

### 4.2. Practical Implication

The initial results allow us to encourage the use of programs and volunteering that help in the development of prosocial behaviours both with children and adults. Furthermore, the use of valid instruments for the measurements of all variables to be investigated is pointed out.

### 4.3. Future Research Lines

First, more research is needed to study the relationship between prosocial behaviours and addictions in different countries and different social, gender, and age groups as well as to isolate different addictive substances and behaviours. Second, the exploration of the relationships between these constructs was focused exclusively on linear dependency, completely ignoring the possible and reasonable existence of non-linear associations. Therefore, it may be necessary to introduce procedures sensitive to non-linear associations. Third, there is also a need for research to use standardised, valid, and reliable measurement and evaluation instruments to ensure that what is being studied is properly analysed. In this sense, the incorporation of minimum evidence of validity of the instruments in substantive investigations (i.e., non-psychometric) is required, especially the evidence of internal structure and reliability.

## 5. Conclusions

The review shows that the relationship between prosocial behaviour and addiction problems has not yet been thoroughly researched, and although the results seem to indicate that it does exist regardless of the type of addiction, further analyses are necessary in order to reach more concrete conclusions. The strength of the relationship ranged from moderate to small. Evidence for relationships was obtained predominantly from cross-sectional research designs. There was a low proportion of studies that induced the validity of the instruments used, and this could generate measurement bias. Researchers studying this topic are encouraged to use validated instruments and tests or objective medical tests in the case of addictions as well as to include groups of participants with specific addiction problems in their research.

## Figures and Tables

**Figure 1 healthcare-10-00074-f001:**
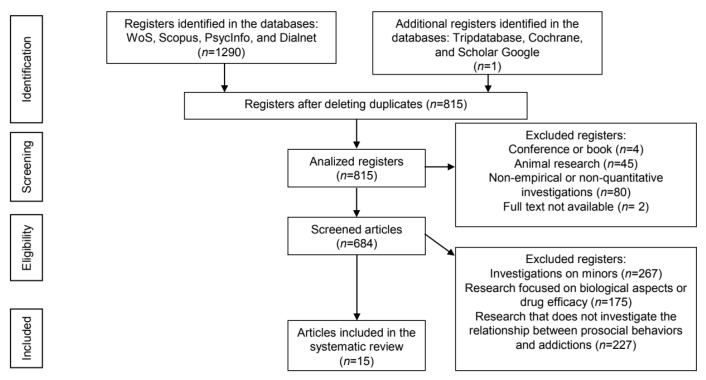
Flowchart of the selection and screening process of the systematic review articles according to the PRISMA method.

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
