# Peer review of "Relationship between Prosocial Behaviours and Addiction Problems: A Systematic Review"

_healthcare, 2021, doi:10.3390/healthcare10010074_

Round 1

Reviewer 1 Report

In this paper the authors review 15 papers dealing with addictions and prosocial behaviours. Most found a negative relationship, although null findings and positive relationships were reported by a small number of studies. The authors conclude that this is an understudied area. 

Whilst I agree that this is an understudied area, I am not sure this review provides insights that add greatly to what is already known. It is already clear that this area is not studied much and it is not surprising, theoretically or empirically, that addiction is negatively linked with prosocial behaviour. What would have added more to knowledge would have been some sort of clear idea as to which way the causal arrow goes – do addictions reduce prosocial behaviour or are antisocial people more likely to have an addiction? However, this issue, whilst touched on briefly, was not dealt with in analyses or discussed in detail.  

This paper has also has some grammatical errors and would benefit from a thorough proof reading to enhance clarity and grammatical correctness. 

The abstract does not, for me, adequately encapsulate the key findings of the study. 

Page 1. Strictly speaking antisocial behaviours are just that - behaviours – and not negative personality traits. Negative traits may or may not lead to antisocial behaviours.  

Page 2. The search terms used did not seem very comprehensive. 

The abstract says 11 articles were selected, the PRISMA diagram and the Table show 15.  

The analysis was fairly shallow in places, simply noting the percentage of studies dealing with a particular issue. In my view a more thorough and detailed analysis would have strengthened the paper. 

I found the valence a little confusing in places, possible because studies were correlational, but it wasn’t always clear whether the finding was that a certain prosocial behaviour was linked to lower addiction or whether addiction was linked to lower levels of prosocial behaviour. Clarity around causality throughout would have been helpful. 

Page 10. It was noted that “20.68% (n = 6) of the results in the investigation between addictive behaviours and prosocial behaviours found a negative relationship; of this proportion, 17.24% (n = 5) found that both variables correlate negatively and the remaining 13.79% (n = 4)”. I found this a little confusing – did 5 out of 6 find a negative correlation (80+%)? How were four left over if there were just 6 studies to start? This section needs to be rewritten for clarity. 

Pages 10-11. It is noted that most studies did not use validated tests or medical analysis. However there is no analysis of these aspects in the study. The Table and analyses should make an empirically derived judgement about the quality of each measure in each study to provide evidence for this assertion. This would also provide a better basis for the later discussion point that some studies which found no relationship or a positive relationship used poorer measures. 

Page 11. I am not sure it is unbalanced or problematic to have 55% of studies on substances and 45% on behavioural addictions. 

Overall this paper, for me, did not add enough to knowledge in this field to justify publication. It is possible that a substantial revision with additional analyses may provide a stronger contribution.  

Author Response

It is already clear that this area is not studied much and it is not surprising, theoretically or empirically, that addiction is negatively linked with prosocial behaviour. What would have added more to knowledge would have been some sort of clear idea as to which way the causal arrow goes – do addictions reduce prosocial behaviour or are antisocial people more likely to have an addiction? However, this issue, whilst touched on briefly, was not dealt with in analyses or discussed in detail.

We believe that the modifications made will solve this problem.

This paper has also has some grammatical errors and would benefit from a thorough proof reading to enhance clarity and grammatical correctness.

Thanks for your input, the manuscript has been reviewed by a native translator to fix any errors.

The abstract does not, for me, adequately encapsulate the key findings of the study.

Page 1. Strictly speaking antisocial behaviours are just that - behaviours – and not negative personality traits. Negative traits may or may not lead to antisocial behaviours.

Page 2. The search terms used did not seem very comprehensive.

We have tried to fix these errors; hopefully you are satisfied with the changes.

The abstract says 11 articles were selected, the PRISMA diagram and the Table show 15.

It was an error in the abstract that has already been fixed. Thanks.

The analysis was fairly shallow in places, simply noting the percentage of studies dealing with a particular issue. In my view a more thorough and detailed analysis would have strengthened the paper. 2

We have tried to fix these errors. We hope you are satisfied.

I found the valence a little confusing in places, possible because studies were correlational, but it wasn’t always clear whether the finding was that a certain prosocial behaviour was linked to lower addiction or whether addiction was linked to lower levels of prosocial behaviour. Clarity around causality throughout would have been helpful.

We have tried to approach it differently to improve the manuscript

Page 10. It was noted that “20.68% (n = 6) of the results in the investigation between addictive behaviours and prosocial behaviours found a negative relationship; of this proportion, 17.24% (n = 5) found that both variables correlate negatively and the remaining 13.79% (n = 4)”. I found this a little confusing – did 5 out of 6 find a negative correlation (80+%)? How were four left over if there were just 6 studies to start? This section needs to be rewritten for clarity.

We think that it’s already corrected.

Pages 10-11. It is noted that most studies did not use validated tests or medical analysis. However there is no analysis of these aspects in the study. The Table and analyses should make an empirically derived judgement about the quality of each measure in each study to provide evidence for this assertion. This would also provide a better basis for the later discussion point that some studies which found no relationship or a positive relationship used poorer measures.

Finally, we do not include this analysis because it seems to us that it would be more interesting to carry it out in a broader investigation than to include more articles. Therefore, we will consider your comments for future research.

Page 11. I am not sure it is unbalanced or problematic to have 55% of studies on substances and 45% on behavioural addictions.

Se ha corregido este error.

Reviewer 2 Report

  1. Abstract: "The objective of this research is to carry out a systematic review of the scientific literature that has studied this relationship from 1900 to 2020" - 1900 or 1990?
  2. "The search was made in titles, abstracts, and keywords, and no restriction was included for years, so it covers from 1900 to July 2021" - 1900 or 1990?
  3. "First, all the articles analysed were published in the last 20 years, 80% in the last 10 years and 53.3% of them belonging to the last 5 years (taking into account that at the time of this review the year 2021 has not yet been completed and may increase the number of articles during that year)" - 1900 or 1990?
  4. The article does not contribute much to science.
  5. Some conclusions are too obvious.
  6. The method of analysis is described too generally.

Author Response

  1. scientific literature that has studied this relationship from 1900 to 2020" - 1900 or 1990?

ANSWER

The dates indicated are correct. The review was carried out without including limits by years in the search. However, no articles prior to 1991 were found.

  1. "The search was made in titles, abstracts, and keywords, and no restriction was included for years, so it covers from 1900 to July 2021" - 1900 or 1990?

ANSWER

The dates indicated are correct. The review was carried out without including limits by years in the search. However, no articles prior to 1991 were found. The method has been nuanced to avoid misinterpretation.

  1. "First, all the articles analysed were published in the last 20 years, 80% in the last 10 years and 53.3% of them belonging to the last 5 years (taking into account that at the time of this review the year 2021 has not yet been completed and may increase the number of articles during that year)" - 1900 or 1990?

ANSWER

The dates indicated are correct. The review was carried out without including limits by years in the search. However, no articles prior to 1991 were found. The method has been nuanced to avoid misinterpretation.

  1. The article does not contribute much to science.

ANSWER

We hope that with the modifications made you consider that this problem has been solved.

  1. Some conclusions are too obvious.

ANSWER

In the updated manuscript, we have tried to correct the deficiencies that you indicated.

  1. The method of analysis is described too generally.

ANSWER

We consider that the changes made to the manuscript have strengthened the description of the method.

Reviewer 3 Report

The authors have developed a manuscript with the objective of conducting a systematic review between addictions in adults and prosocial behaviors.
The manuscript has quite a few formatting deficiencies and your search cannot be reproduced.

More specifically:

INTRODUCTION

  • The introduction should not be subdivided into other sections. There must be a fluent speech.
  • The research objective is poorly stated. The goal cannot be "to conduct a systematic review." The verb must be a concrete action in relation to addictions among adults.
  • The authors do not explain well the concepts necessary to understand the research problem. For example, in the inclusion criteria they distinguish between “prosocial behaviors”, “prosocial Jobs”, “prosocial attachments” or “prosocial environments”.

METHODS

  • Reference 28 referring to the PRISMA declaration is obsolete. There have been updates since then. There are references with an update of 2020.

Search strategy:

  • It is not understood why the authors start looking in a metasearch (WOS) together with some databases, and then search in Pubmed and finally in Google Scholar. The latter is a meta search engine as well.
  • The authors have misused quotation marks. The search cannot be played as is. For example, depending on whether they are removed from where it is left over, in academic google it varies from 7260 to 4940 results.
  • Has the search equation used descriptors in English and Spanish? How many results came out in each case? Authors must specify.
  • The dialnet database distinguishes between singular and plural, masculine and feminine. Have the authors taken this into account if they search with descriptors in Spanish? If they used truncation they must specify the search equation in Spanish.

Inclusion / exclusion criteria:

  • The authors say they only used empirical articles:
    o Were doctoral theses included?
    o Were editorials included?
    o Were clinical case studies included?

Flowchart:

  • The figures in the diagram do not correspond to what the authors have described. For example, in Dialnet, depending on the quotation marks, 7260 or 4940 results appear. Authors should describe the figures in each database more fully.
  • In addition, the authors refer to a complementary search in Pubmed and it does not appear in the diagram. The equation in Pubmed yields 662 results. Authors should explain why they did not analyze these results.

Evaluation of the quality of the study:

  • What tool have the authors used to assess the quality of the studies?

RESULTS

  • Authors must order table 1 by order of appearance in the text, not by alphabetical order. They have mixed APA citation regulations with Vancouver. Alphabetical order is not used in Vancouver.
  • Also the figures do not match. The authors say that there are 15 articles and they should be references 29 to 43. In the table they go up to 56 and figures are skipped. There is disorder in the text.
  • Authors should include the study design in Table 1.

DISCUSSION

  • The authors have used references to the “results” in the “discussion”. That is, they have not justified their results with other research. This is not adequated.

REFERENCES

  • There are many misprints in the references. For example there are full magazine titles and other abbreviated ones; DOIs are set differently; etc.

Author Response

The authors have developed a manuscript with the objective of conducting a systematic review between addictions in adults and prosocial behaviors.
The manuscript has quite a few formatting deficiencies and your search cannot be reproduced.

More specifically:

INTRODUCTION

  • The introduction should not be subdivided into other sections. There must be a fluent speech.

ANSWER

We have modified the structure of the introduction to solve this problem. Thank you for your recommendation.

  • The research objective is poorly stated. The goal cannot be "to conduct a systematic review." The verb must be a concrete action in relation to addictions among adults.

ANSWER

We have changed the wording of the objective. Thanks for pointing this out to us.

  • The authors do not explain well the concepts necessary to understand the research problem. For example, in the inclusion criteria they distinguish between “prosocial behaviors”, “prosocial Jobs”, “prosocial attachments” or “prosocial environments”.

ANSWER

The objective was to search for "prosocial behavior", but this term appears in the scientific literature with many synonyms such as prosociality .... In the search equation the word prosocial * was put, since with this the root of the word prosocial. In the screening, eligibility and inclusion, all those articles that did not evaluate prosocial behavior were eliminated.

METHODS

  • Reference 28 referring to the PRISMA declaration is obsolete. There have been updates since then. There are references with an update of 2020.

Updated reference added.

Search strategy:

  • It is not understood why the authors start looking in a metasearch (WOS) together with some databases, and then search in Pubmed and finally in Google Scholar. The latter is a meta search engine as well.

There has been an error interpreting the manuscript. What we wanted to say is that in the first place a previous search was carried out to rule out the existence of similar articles. After this, the search was carried out simultaneously in all the databases and metasearch indicated (using a bibliography manager, Refwroks, to combine all the documents and work on them at the same time).

  • The authors have misused quotation marks. The search cannot be played as is. For example, depending on whether they are removed from where it is left over, in academic google it varies from 7260 to 4940 results.

There was a mistake in the wording of the search equation. It has already been corrected.

  • Has the search equation used descriptors in English and Spanish? How many results came out in each case? Authors must specify.

The search equation was carried out in English. The only exception was Dialnet, since it only works in Spanish. Among other things, this is what we mean when we say that the search equation was adapted to the characteristics of each database.

  • The dialnet database distinguishes between singular and plural, masculine and feminine. Have the authors taken this into account if they search with descriptors in Spanish? If they used truncation they must specify the search equation in Spanish.

As indicated in the manuscript, we adapt the search equation to the characteristics of each database.

  • The authors say they only used empirical articles:
    o Were doctoral theses included?
    o Were editorials included?
    o Were clinical case studies included?

We have included this information in the method. Thanks.

  • The figures in the diagram do not correspond to what the authors have described. For example, in Dialnet, depending on the quotation marks, 7260 or 4940 results appear. Authors should describe the figures in each database more fully.

There was a mistake in the wording of the search equation. It has already been corrected.

  • In addition, the authors refer to a complementary search in Pubmed and it does not appear in the diagram. The equation in Pubmed yields 662 results. Authors should explain why they did not analyze these results.

Pubmed database was not worked. It was an error in the manuscript; has already been corrected.

  • What tool have the authors used to assess the quality of the studies?

No tools were used for this purpose. We will consider your suggestion for future research.

RESULTS

  • Authors must order table 1 by order of appearance in the text, not by alphabetical order. They have mixed APA citation regulations with Vancouver. Alphabetical order is not used in Vancouver.

In this case, as these were references that had not yet appeared in the text previously, we decided to present them in the table in alphabetical order.

  • Also the figures do not match. The authors say that there are 15 articles and they should be references 29 to 43. In the table they go up to 56 and figures are skipped. There is disorder in the text.

In this case, as these were references that had not yet appeared in the text previously, we decided to present them in the table in alphabetical order.

  • Authors should include the study design in Table 1.

This section has been incorporated following your suggestion.

DISCUSSION

  • The authors have used references to the “results” in the “discussion”. That is, they have not justified their results with other research. This is not adequated.

We hope that the modifications made have solved this problem.

REFERENCES

  • There are many misprints in the references. For example there are full magazine titles and other

We think that it’s already solverd.

Round 2

Reviewer 2 Report

No comments.

Author Response

Revisor 2:

English has been checked throughout the document by a native translator.

Reviewer 3 Report

Dear authors, I appreciate your effort to improve the quality of this manuscript. However, there still are some issues that should be taken into account before considering this work for publication.

There is still a lack of information related to the search process. Please aclare. In this sense the authors indicates that the search equation used descriptors in English and Spanish, but they use the Dialnet database that not produce the same results in Spanish depending on the gender and number of the used words for this search. Please provide detailed information about this topic.

The authors answer to this reviewer that no tools were used to assess the quality of the studies. In this sense They should use, for example, the OCEBM tool that is adequate to their review. It must to be applied.

I still think that those references that had not yet appeared in text previously shouldn’t be presented in the table. Here, in the table, the authors should review the English language within the table because, for example, when they write ‘Transversal’ it is probably they wanted to write ‘cross-sectional’.

Finally, the authors write that they solve the misprints in the references but it is very difficult to review this fact if they don’t highlight these changes.

Author Response

Revisor 3:

There is still a lack of information related to the search process. Please aclare. In this sense the authors indicates that the search equation used descriptors in English and Spanish, but they use the Dialnet database that not produce the same results in Spanish depending on the gender and number of the used words for this search. Please provide detailed information about this topic.

The search equation used in Dialnet has been specified to avoid errors.

The authors answer to this reviewer that no tools were used to assess the quality of the studies. In this sense They should use, for example, the OCEBM tool that is adequate to their review. It must to be applied.

We have tried to use the OCEBM tool for our manuscript, but when dealing with psychological constructs we have not found good examples to help us apply this tool.

I still think that those references that had not yet appeared in text previously shouldn’t be presented in the table. Here, in the table, the authors should review the English language within the table because, for example, when they write ‘Transversal’ it is probably they wanted to write ‘cross-sectional’.

We have corrected these and other errors in the English with the help of a native translator.
